# Coping Strategies by University Students in Response to COVID-19: Differences between Community and Clinical Groups

**DOI:** 10.3390/jcm10112499

**Published:** 2021-06-05

**Authors:** Víctor-María López-Ramos, Benito León-del-Barco, Santiago Mendo-Lázaro, María-Isabel Polo-del-Río

**Affiliations:** Psychological Counseling Services, Faculty of Teacher Training College, University of Extremadura, 10071 Cáceres, Spain; vmlopez@unex.es (V.-M.L.-R.); mabelpdrio@unex.es (M.-I.P.-d.-R.)

**Keywords:** pandemic, lockdown, mental health, anxiety, stress, psychological coping

## Abstract

Last year, the COVID-19 pandemic had severe consequences on the health and well-being of millions of people. Different studies try to identify the main effects that the crisis and several lockdowns have had on the citizens’ mental health. This research analyses the coping strategies generated by students from a community group and a clinical group in response to this crisis, using the Coping Responses Inventory—Adult Form (CRI-A) by Moos with a sample of 1074 students of Universidad de Extremadura. Multivariate analysis and receiver operating characteristic curve analysis have been carried out, revealing, amongst other things, a greater predisposition of the clinical sample towards factors such as seeking guidance and support, cognitive avoidance or emotional discharge. Results show that students with prior mental health problems perform an unhealthy coping response based on avoidance strategies. This group of students suffers a double source of distress and anxiety, one derived from their prior psychopathologic problems and the stress of the lockdown and another one originating from an inefficient coping response, which makes coping strategies raise levels of distress and anxiety.

## 1. Introduction

On 11 March 2020, the World Health Organisation (WHO) declared COVID-19 (acronym for “coronavirus disease 2019”) a pandemic due to the high morbidity and mortality registered since this novel coronavirus was first detected in the city of Wuhan (China) in December 2019. The health, economic and social effects of the pandemic are currently extremely severe, and there is still no confirmation of how long they will last. Consequently, the Government of Spain approved a Royal Decree (RD 463/2020, on 14 March) that declared a state of alarm in order to manage the sanitary crisis caused by the pandemic. During the state of alarm, movement had to be individual and limited to first-need activities or commuting to workplace; passenger transport options were radically reduced; cultural, artistic and sporting venues were closed; working from home became a priority and face-to-face education was suspended at all levels, favouring online education.

The lockdown experienced by the Spanish society, as well as almost every country nearby, was an extraordinary situation as it was unfamiliar and has proved to have a strong impact on the psychological well-being of citizens, with various sources of stress.

A first study on the psychological impact of the COVID-19 quarantine in China [1] showed that psychosocial stress and the loss of habits and routines are the two main factors affecting physical and mental well-being during a period of confinement such as the one we have experienced. Studies on situations of risk, conflict and emergencies allow us to synthetise that the main variables implied in psychological impact are the following: fear of contracting diseases, feelings of frustration and boredom, not being able to meet basic needs and the lack of information and clear action guidelines [2], or the presence of previous mental health pathologies or economic hardships [3]. Additionally, stigma and social rejection of people infected or exposed to the disease are potential triggers for having difficulties when adapting to the situation [2]. The impact degree depends on several factors. According to Sprang and Silman’s research [4], people who have already experienced a quarantine during a pandemic are more likely to develop acute stress, adaptation and pain disorders (30 percent of them presenting criteria of post-traumatic stress disorder).

Nonetheless, there is still scant evidence about the immediate psychological impact of the consequences of the pandemic on the general population, mainly through research studies carried out with Chinese individuals. In a first study with a sample of 1210 people, 53% valued the psychological impact of the situation as moderate to severe, 16% referred moderate to severe depression symptoms, 28% referred moderate to severe anxiety symptoms and 8% declared moderate to severe levels of stress. For 75% of the individuals studied, the main concern was their relatives becoming infected with the disease [1]. Another research carried out with inhabitants of Wuhan and neighbouring towns showed a prevalence of 7% for symptoms of post-traumatic stress [5]. Moreover, the same group considered a wider sample of 2091 people and got a prevalence of 4.6% for symptoms of acute post-traumatic stress one month after the COVID-19 outbreak [6].

The social–cultural and psychosocial reality in Spain may have elements of connection with the results obtained in these and other international research studies, although we could also anticipate important cultural, social and health particularities. During the months of April, May and June 2020, several surveys were carried out, most of them with reduced and not very representative samples, as well as polls that compiled basically descriptive data. Broadly, conclusive results on these kinds of studies are yet to be published, although we do have some works that are starting to offer valuable data, such as the study developed by Ozmiz-Etxeberria et al. [7] with a sample of 976 people, collected in Northern Spain, which indicates that severe and extremely severe levels of stress, anxiety and depression found in this sample were less than those collected in the study carried out in China by Wang et al. [1]. This is probably due to the fact that Spain had broader access to information about the virus since it got here one and a half month later. Otherwise, it is noteworthy that the study by Ozmiz-Etxeberria et al. [7] found higher averages for stress, anxiety and depression in people aged 18–25 years old, followed by those aged 26–60, the average in the three dimensions being lower in those over 60. This may imply that the younger group of the study mainly comprised students and that the stress generated was increased by added stress due to the need to adapt to the new educational context without on-site classes. Another study concluded that 89% of a sample composed by children presented behavioural or emotional alterations as a consequence of lockdown [8]. In this sense, the fact of resuming healthy routines and habits once lockdown was over, together with getting healthy support, must allow the affected individuals to recover to normal functioning [9]. Nevertheless, the need of some type of psychological support after a period of lockdown is to be expected, especially amongst those who presented previous psychological conditions, development disorders or other psychopathologies [10].

It is evident that the impact of lockdown measurements on the general population between the months of March and June, and even the consequences of the restrictive measures prevailing after the state of alarm, have differently affected the various populational sectors that compose Spain’s social structure. We consider that, for university students, this impact requires a more detailed study since apart from being young—18 to 25—they were also forced to end the school year 2019–2020 in an online modality and to face final exams also virtually. Determining the extent of consequences for university students would significantly aid the proposals of improvement related to psychological orientation and counselling in this context, and it would also allow the definition of proceedings aimed to manage stress and anxiety in universities, especially in situations of crisis, with a particular approach to individuals with prior mental health issues or disorders.

The unusual, unpredictable nature of the lockdown imposed in Spain to control the spread of COVID-19 gives us an opportunity to reflect on the personalised support models that should be put in place for university students, especially in risk and conflict situations. In this regard, echoing Balluerka et al. [10], individuals who are predisposed to certain problems or who have displayed psychopathological symptoms in the past are likely to be at greater risk of these symptoms reappearing after lockdown. Therefore, in line with Balluerka et al. [10], two types of psychological effects are anticipated to have emerged during and after the lockdown among our sample of university students:Specific effects that are directly linked to one or more stimuli caused by the COVID-19 pandemic: family and personal circumstances leading to moderate to severe suffering as a result of high levels of stress and emotional disturbance, with medical, social and economic repercussions.Unspecific effects that are very difficult to attribute to a specific trigger beyond the contextual changes arising as a result of the pandemic and lockdown, which are linked to intense worry, fear of contagion, pessimism about the future, sense of vulnerability, uneasiness in the face of uncertainty, etc.

Situations such as the one generated by the COVID-19 pandemic force us more precisely define the conceptualisation of coping strategies as a fundamental element for adaptations made by people in every diverse critical situation they may face [11]. In this way, coping must be approached as a stabilising variable that helps people to keep their psychosocial adaptation in moments of high stress levels [12,13]. A number of studies have concluded that coping strategies focused on the problem itself reduce psychological distress, whereas strategies based on emotions increase it [14]. In this sense, active skills appear to be associated with health, while avoidance ones are linked to the development of a range of diseases [15,16].

The characteristics and nature of the events faced by people influence both the availability and mobilisation of resources, as well as the coping strategies for those [17] (Moos, 1993). From the coping model raised by this author, the characteristics specific to a crisis and the evaluation of the situation carried out by an individual contextualise the response choice specific to coping [11]. It has been observed that stressful situations tend to promote a higher amount of behavioural active coping responses, while those due to interpersonal relationships generate a higher amount of coping focused on emotion [18]. In this sense, it seems to be demonstrated that the more negative life situations and chronic stress sources are present, the less employment of responses focused on approach to the problem there are, and higher the use of responses focused on avoidance is [19]. In relation with evaluation of stressful life situations, it has been checked that, when stressors are evaluated as a challenge, they tend to provoke coping responses focused on approach more than on avoidance. This means that type, severeness and evaluation performed in moments of crisis influence the coping strategies used, which proves the need and pertinence of the interconnection established between the different coping responses and concretion of the situation [19].

Another element of key importance is the degree of controllability of the stressful situation itself. The perceived controllability of the stressful stimulus has an impact on the type of strategy used and on its effectivity for lowering the stress level [20]. In this sense, Moos and Schaefer [13] propose that coping responses focused on approach should be more effective in situations that are regarded as changeable and controllable. In this regard, studies by Mikulic and Crespi [21,22] carried out with people in situations of deprivation of liberty found a prevalence of coping responses focused on avoidance, which are related to the perception of not being able to operate in the imprisonment situation that generates discomfort because of the multiplicity of variables beyond the control of the individuals.

Along these lines, the Coping Responses Inventory—Adult Form (CRI-A) by Moos becomes an important tool for the study of coping responses in an adult population thanks to its psychometric inputs in multiple sociocultural contexts [11].

### The Present Study

This study analyses the coping strategies adopted by university students in response to the COVID-19 pandemic. To do this, it uses the Coping Responses Inventory—Adult Form (CRI-A) by Moos [17].

Drawing on previous studies, the aim is to compare a community group with a clinical group to identify discriminative coping responses displayed by students in response to the Spanish lockdown lasting several months. Two key variables have been established for the study: the group to which the individual belongs, either community or clinical, and the coping response adopted by the individual according to the typology set out in the CRI-A (logical analysis; positive reappraisal; seeking guidance and support; problem solving; cognitive avoidance; acceptance or resignation; seeking alternative rewards; and emotional discharge).

## 2. Materials and Methods

### 2.1. Participants

In order to be included in the study, participants had to be enrolled at the University of Extremadura during the 2019–2020 academic year. The sample was made up of 1074 students. The average age was 23.07 years old (SD = 5.28; range 18–37); 56.4% (n = 606) were female and 43.6% (n = 468) were male. In all, 16.8% of the students were in their first year, 18.2% in their second year, 25.9% in their third year, 21.2% in their fourth year and 17.9% in their fifth, sixth or master’s year on different degree courses at the University of Extremadura. The number of participants was calculated on the basis of the 20,000 students enrolled in the 2019–2020 academic year, considering a sample error of 3% and a confidence level of 96%. Students were selected from the faculties at the University of Extremadura using multistage cluster sampling and random selection by degree course and year of study. The sample from the psychopathological clinical population was made up of 135 individuals: 60% were male and 40% were female. The clinical group comprised students diagnosed with mental health disorders, predominantly anxiety and depression, by public and private mental health services. 

Both the clinical and the community group were equivalent in age *t* (1063) = −0.478, *p* = 0.633, gender χ^2^(1) = 0.197, *p* = 0.657, and year χ^2^(6) = 5.940, *p* = 0.460.

### 2.2. Instruments

The adaptation by Kirchner and Forns [23] of the Coping Responses Inventory—Adult Form (CRI-A) by Moos [17] was used. The inventory consists of 48 items on a Likert scale of 0 to 3 (0 = never; 1 = rarely, 2 = occasionally; 3 = very frequently) evaluating eight factors (six items per factor; minimum score = 0; maximum score = 18) or coping responses: (1) Logical analysis (LA): cognitive attempts to understand and mentally prepare to cope with a stressor and its consequences. (2) Positive reappraisal (PR): cognitive attempts to construct and restructure a problem in a positive manner while accepting the reality of the situation. (3) Seeking guidance and support (SG): behavioural attempts to seek out information, support and guidance. (4) Problem solving (PS): behavioural attempts to carry out actions leading to the root of the problem. (5) Cognitive avoidance (CA): cognitive attempts to avoid thinking about the problem realistically. (6) Acceptance or resignation (AR): cognitive attempts to respond to the problem by accepting it. (7) Seeking alternative rewards (SR): behavioural attempts to engage in alternative activities and create new sources of satisfaction. (8) Emotional discharge (ED): behavioural attempts to reduce tension by expressing negative feelings. The LA, PR, CA and AR strategies relate to the cognitive dimension of coping, whereas SG, PS, SR and ED relate to the behavioural dimension. The LA, PR, SG and PS coping responses are considered approach strategies, while the CA, AR, SR and ED responses are considered avoidance strategies.

Kirchner and Forns [23] reviewed and validated the psychometric properties of the questionnaire in its Spanish version in line with the coping responses proposed by Moos [17], who notes that individuals tend to respond to a specific situation in either a cognitive or behavioural manner, with a focus on approach or avoidance strategies [24]. Table 1 summarises the response types in relation to the eight factors of the CRI-A:

In relation to reliability, values are similar to those from the CRIA-A adaptation study [23]. Cronbach’s alpha’s values (α) of all eight strategies were between 0.61 and 0.73, presenting a lower value than expected for the AL factor (α = 0.61). All four dimensions—cognitive (α = 0.72), behavioural (α > 0.71), approach (α = 0.83) and avoidance (α = 0.71)—obtained acceptable values.

### 2.3. Procedure

The CRI-A was administered online using the Google Forms application (a Google Drive tool, for the individuals of both the community and the clinical group. In accordance with the ethical guidelines issued by the American Psychological Association (APA, 2009), all participants gave their informed consent before completing the questionnaires. The questionnaires were anonymous, guaranteeing the confidentiality of the data obtained and the exclusive use of these data for research purposes. Data were collected between 10 May and 10 June 2020, during home lockdown decreed by Spain’s government (from 14 March to 20 June 2020). When this research was carried out, in Spain there were 242,280 officially confirmed COVID-19 cases and 27,136 people had died from the disease.

At the time of the data collection, the clinical group was formed by 113 people who were being treated or had been previously treated at the Psychological Care Unit (UNAP, for its acronym in Spanish), reporting to the Research Group in Social and Personality Evolutive Psychology (GIPES-UEx). This counselling is available on request by the students and does not aim to be a clinical psychological intervention but an action based on psychological counselling and on strategy training and personal advice oriented towards studying the efficiency of this counselling and towards the improvement of their interpersonal relationships, taking into account the disorders and alterations suffered.

In this group, 46 individuals had a previous diagnosis performed by external professionals not related to UNAP, from both public and private mental health services, and 67 had been assessed within UNAP. The most diagnosed disorder was related to general anxiety episodes (52% of individuals from the clinical group), as wells as fears or specific phobias with continuous concern, exaggerated or unrealistic (43%). Diagnoses of depression (32%) and obsessive–compulsive disorder (14%) were also remarkable; as for the clinical assessments carried out by UNAP, the most relevant findings were those related to sleep (75%) and eating (60%) disorders, and also alterations in intimate relationships and in sexual behaviour and health, with special emphasis on aspects such as communication between intimate partners (26%), emotional climate in the relationship (24%) or a decreasingly satisfactory sex life (22%).

The study was conducted according to the guidelines of the Declaration of Helsinki and was approved by the Bioethics and Biosafety Committee of the University of Extremadura (21/001/UAP). 

### 2.4. Data Analysis

SPSS 21.0 was used to perform statistical analysis on the collected data (IBM Corp., New York, NY, USA, 2012). The reliability of the instruments used was calculated using Cronbach’s alpha. After checking the assumptions of normality and homoscedasticity, a multivariate analysis (MANOVA) and receiver operating characteristic curve analysis (ROC) were carried out.

## 3. Results

Firstly, multivariate comparisons of the mean scores for the CRI-A factors were carried out by group (community/clinical) and gender, as well as for the interaction between the two variables (Table 2).

The multivariate analysis (MANOVA) gave a significant main effect by group (community/clinical), Wilks’ λ = 0.901, *F* (8, 1063) = 14.594 *p* < 0.001, *η* = 0.099, and by gender, Wilks’ λ = 0.982, *F* (8, 1063) = 2.372, *p* = 0.016, *η* = 0.018), although no significant main effect was found for the group–gender interaction, Wilks’ λ = 0.988, *F* (8, 1063) = 1.555, *p* = 0.134, *η* = 0.012.

As for the univariate contrasts, the tests for between-individual effects (Table 2) revealed significantly higher scores in the clinical group (*p* < 0.05) than in the community group in the seeking guidance and support, problem solving, cognitive avoidance, acceptance or resignation, seeking alternative rewards and emotional discharge factors. Male participants obtained significantly higher scores (*p* = 0.014) in the logical analysis factor than female participants did.

Multivariate comparison of the average scores in the CRI-A dimensions was carried out by group (community/clinical) and gender, as well as for the interaction between the two variables (Table 3).

The multivariate analysis (MANOVA) gave a significant main effect by group (community/clinical), Wilks’ λ = 0.905, *F* (8, 1063) = 33.095 *p* < 0.001, *η* = 0.085, and by group–gender interaction, Wilks’ λ = 0.990, *F*(8, 1063) = 3.430, *p* = 0.017, *η* = 0.010, although no significant main effect was found for gender, Wilks’ λ = 0.998, *F* (8, 1063) = 0.593, *p* = 0.620, *η* = 0.002.

As for the univariate contrasts, the tests for between-subject effects (Table 3) show that the clinical group obtained significantly higher scores (*p* ≤ 0.001) than the community group in the cognitive, behavioural and avoidance dimensions. They also revealed a significant group–gender interaction (*p* < 0.05) in the behavioural and avoidance dimensions.

The pairwise comparisons for interaction effects showed, on the one hand, that the difference between the groups (clinical/community) in the behavioural dimension was only significant (*p* = 0.002) between men, and, on the other hand, that men obtained significantly higher scores in the behavioural (*p* = 0.013) and avoidance (*p* ≤ 0.001) dimensions in the community sample.

To analyse differences in the most commonly adopted strategies between the community and clinical groups, a repeated-measures ANOVA was performed between the scores for the cognitive, behavioural, approach and avoidance dimensions for the community and clinical groups (Table 4).

The repeated-measures analysis showed significant differences (*p* < 0.05) between the scores obtained in the four dimensions of the CRI-A in both groups (Table 4). The pairwise comparisons indicated the following: (1) In the community group, significantly higher scores (*p* < 0.001) were obtained in the cognitive and approach dimensions than in the behavioural and avoidance dimensions, with the score (*p* < 0.001) for the approach dimension being significantly higher than for the cognitive dimension. (2) In the clinical group, significantly higher scores (*p* < 0.05) were obtained in the cognitive and avoidance dimensions than in the behavioural and approach dimensions.

In order to evaluate the discriminative accuracy of the coping responses, a receiver operating characteristic curve analysis (ROC) was carried out to identify the cut-off points for the scores in the dimensions of the CRI-A (cognitive, behavioural, approach, avoidance), after which behavioural disorders become more likely.

In the nonparametric ROC analysis (Figure 1), the area beneath the curve for the cognitive dimension is 0.609 (*p* < 0.001; confidence interval 95%; min. = 0.555; max. = 0.622), the area beneath the curve for the behavioural dimension is 0.582 (*p* = 0.002; confidence interval 95%; min. = 0.529; max. = 0.635), the area beneath the curve for the approach dimension is 0.496 (*p* = 0.875; confidence interval 95%; min. = 0.444; max. = 0.547) and provides no significant information, and finally, the area beneath the curve for the avoidance dimension is 0.724 (*p* < 0.001; confidence interval 95%; min. = 0.683; max. = 0.765).

Table 5 shows the cut-off points maximising both sensitivity and specificity and the cut-off points maximising sensitivity and specificity for the dimensions providing significant information (*p* ≤ 0.05).

With regard to the presence of psychopathological disorders, a score of ≥42.5 in the cognitive dimension maximised both sensitivity (58%) and specificity (58%) (Youden’s index = 0.159), a score of 41.5 maximised sensitivity (60%) while maintaining specificity higher than random, and a score of 43 maximised specificity (60%) while maintaining sensitivity higher than random. In the behavioural dimension, a score of ≥39.5 maximised both sensitivity (53%) and specificity (59%) (Youden’s index = 0.124), a score of 37.5 maximised sensitivity (56%) while maintaining specificity higher than random, and a score of 40.0 maximised specificity (61%) while maintaining sensitivity higher than random. Finally, in the avoidance dimension, a score of ≥39.5 maximised both sensitivity (64%) and specificity (65%) (Youden’s index = 0.293), a score of 36.0 maximised sensitivity (79%) while maintaining specificity higher than random, and a score of 42.0 maximised specificity (73%) while maintaining sensitivity higher than random.

## 4. Discussion

The lockdown restrictions imposed in Spain and across Europe gave rise to a unique and unprecedented situation that had a serious impact on the population’s psychological well-being, generating a number of sources of stress. The aim of this study was to ascertain discriminative coping responses to lockdown by university students, divided into a community group and a clinical group. 

Firstly, the results of the multivariate comparisons of the average scores for the factors and dimensions of the CRI-A by group (community/clinical) and gender showed that the clinical sample obtained significantly higher scores than the community group in the seeking guidance and support, cognitive avoidance, acceptance or resignation, seeking alternative rewards and emotional discharge factors. The effect size tests indicated a medium–high effect in the cognitive avoidance (*η* = 0.022), acceptance or resignation (*η* = 0.057) and emotional discharge (*η* = 0.063) factors, which are typical of an avoidance response. Coping using cognitive avoidance and acceptance or resignation strategies, as displayed by the clinical group, represents a more cognitive response. Meanwhile, coping using seeking guidance and support and seeking alternative rewards is suggestive of a more behavioural response. The community group obtained significantly higher scores in the problem solving factor.

Secondly, the repeated-measures analysis and the pairwise comparisons corroborated the results set out above, showing significantly higher scores for the clinical group in the cognitive and avoidance dimensions than in the behavioural and approach dimensions, whereas the community group obtained significantly higher scores in the cognitive and approach dimensions than in the behavioural and avoidance dimensions.

In short, the clinical group adopted an unhealthy coping response based on primarily cognitive avoidance strategies (cognitive avoidance and acceptance or resignation). These strategies have a limited effect and are associated with poorer life satisfaction and more severe psychopathological symptoms [25,26]. The clinical group only displayed one healthier, more active strategy: seeking guidance and support. These results raise several questions: (1) Why does the clinical group adopt a pattern of avoidance coping? (2) Why does the clinical group report more extensive use of one approach strategy, seeking guidance and support? (3) What was the influence of gender?

With regard to the first question, the clinical group uses avoidance coping strategies (cognitive avoidance, acceptance or resignation and emotional discharge) and the community group tends to use approach strategies (problem solving). Active or approach strategies appear to be associated with health, while avoidance strategies are linked to the development of a range of diseases [15,16]. It has been demonstrated that the more chronic stressors are present, the less likely individuals will be to adopt approach responses and the more likely they will be to adopt avoidance responses [16]. Other studies show a positive association between avoidance coping strategies and stress, anxiety, anger, sadness and loneliness [27]. The clinical group experienced distress and anxiety as a result of their prior mental health issues and the stress of the lockdown (loss of habits and routines, fear of contagion, concern for classes and exams, etc.).

The clinical group in this study is a sample of a psychopathological clinical population with a large number of pathologies, predominantly anxiety and depression. In general, people with a psychopathological clinical profile (depression, anxiety, eating disorders, addictions, etc.) tend to use ineffective coping strategies and struggle to adopt a healthy coping response [28]. Some studies have shown an association between avoidance strategies and eating disorders [29]. Other longitudinal studies on mood psychopathology found a significant association between depressive symptoms and avoidance coping responses [30,31,32]. In a similar vein, research has shown that lower use and inhibition of problem solving as a coping strategy is a consequence of depressive states [33].

In answer to the second question, the clinical group obtained significantly higher scores than the community group in the seeking guidance and support factor, which we consider an approach strategy as it represents a behavioural effort to manage or address stressors. Guidance and support may be understood as behavioural attempts to seek information, advice or assistance through social relations with people and groups. According to Yu et al. [34], guidance and support can influence people’s physical and mental health as it is beneficial for all individuals and acts to mitigate stress. A number of studies have shown that seeking guidance and support protects people from developing symptoms of depression and anxiety [35,36,37]. Most existing research indicates that guidance and support have a positive impact on psychological well-being and act as a protective factor against stress [34,38,39,40]. Support from friends and family plays a crucial role in helping individuals to manage stressful situations such as infectious disease outbreaks [41]. It is possible that the differences observed between the clinical group and the community group in terms of seeking guidance and support lie in the fact that the clinical group is more accustomed to seeking support from friends and family due to their existing mental health issues.

In relation to the third question, male participants obtained significantly higher scores than female participants in the logical analysis factor. Logical analysis refers to cognitive attempts to cope with a stressful situation and its consequences and is closely linked to problem solving. Mataud [42] studied gender differences in coping strategies in Spain and found that women obtained lower scores than men in more rational coping strategies and higher scores than men in more emotional strategies. These results coincide with the meta-analysis conducted by Tamres et al. [43] and research by Rose and Rudolph [44], which found that women scored higher on emotional strategies, avoidance and seeking support. More recently, several studies have confirmed that women report greater use of the strategy of seeking guidance and support than men [45,46].

Finally, to evaluate the discriminative accuracy of the coping responses, a receiver operating characteristic curve analysis (ROC) was carried out to identify the cut-off scores on the CRI-A dimensions after which the probability of behavioural disorders rises. The results of this analysis showed that the highest scores for sensitivity and specificity were obtained by avoidance coping strategies. Certain coping strategies can increase the risk of psychopathological disorders and psychopathology can also determine the use of a specific type of coping strategy [47].

### Limitations

This study has several limitations. The use of self-reports as a data collection method is an important limitation. Other limitations include the cross-sectional design, which makes it difficult to establish further inferences about the relationship between the study variables. Finally, cultural influences and differences in the measures adopted between countries during lockdown mean that any attempts to extrapolate the results of this study to other contexts should be approached with caution. Although we obtained group data of the participants enrolled at the University of Extremadura, the generalisation of our findings needs to be considered. The results cannot be generalised to other population groups.

## 5. Conclusions

The study of coping strategies adopted by individuals in response to stressful situations is a crucially important topic due to its association with mental health and psychological well-being. Mental health problems are a matter of significant social concern and lead to high economic costs for public health [48]. This study has advanced our understanding of coping strategies and shown that a stressful situation such as lockdown prompts students with existing health problems to adopt an unhealthy coping response based largely on cognitive avoidance strategies (cognitive avoidance and acceptance or resignation). The clinical group experienced a dual spiral of distress and anxiety due to their psychopathological issues and the stress of the lockdown, as well as their ineffective coping responses. In this case, rather than acting as a protective factor, coping strategies became a risk factor that increased levels of distress and anxiety.

In conclusion, it is important to design interventions to ensure that this clinical group develops active, effective coping strategies in response to stressful situations and learns to manage the anxiety caused by the pandemic. Urgent action must be taken to provide this group with medical and social support.

## Figures and Tables

**Figure 1 jcm-10-02499-f001:**
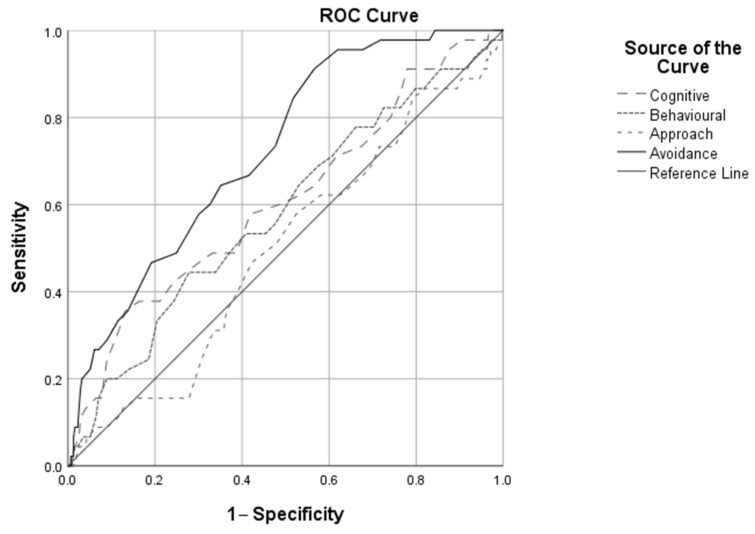
ROC curve for the CRI-A dimensions predicting the presence of psychopathological disorders.

**Table 1 jcm-10-02499-t001:** Coping responses according to Moos (source: Pujada [24]).

		Focus	
		Approach	Avoidance
Method	Cognitive	Logical analysis (LA) Positive reappraisal (PR)	Cognitive avoidance (CA) Acceptance or resignation (AR)
	Behavioural	Seeking guidance and support (SG) Problem solving (PS)	Seeking alternative rewards (SR) Emotional discharge (ED)

**Table 2 jcm-10-02499-t002:** Descriptive analysis for the eight CRI-A factors and univariate analysis by group (community/clinical) and gender.

CRI-A Factors	Sample	Male		Female		Total		Between-Subject Effects Group and Gender		
		M	SD	M	SD	M	SD	F	*p*	*η*
(LA)	Community	10.71	2.79	10.15	3.39	10.28	3.26	1.678	0.195	0.002
	Clinical	11.50	4.02	10.29	3.05	10.56	3.32			
	Total	10.80	2.96	10.16	3.35	10.32	3.27	6.079	0.014	0.006
(PR)	Community	10.72	3.86	11.13	3.81	11.03	3.83	0.951	0.330	0.001
	Clinical	10.40	4.37	10.63	3.36	10.58	3.59			
	Total	10.68	3.92	11.06	3.76	10.97	3.80	0.573	0.449	0.001
(SG)	Community	8.36	3.28	8.85	3.27	8.73	3.28	4.621	0.032	0.004
	Clinical	9.80	5.24	9.03	3.98	9.20	4.28			
	Total	8.53	3.59	8.87	3.37	8.79	3.42	0.141	0.708	0.000
(PS)	Community	11.53	3.62	11.18	3.64	11.27	3.63	6.531	0.011	0.006
	Clinical	10.50	5.13	10.14	3.61	10.22	3.98			
	Total	11.41	3.83	11.05	3.65	11.14	3.69	0.755	0.385	0.001
(CA)	Community	10.09	3.70	10.75	3.29	10.59	3.40	23.881	0.000	0.022
	Clinical	12.60	4.41	11.91	3.14	12.07	3.45			
	Total	10.39	3.86	10.90	3.29	10.78	3.44	0.001	0.971	0.000
(AR)	Community	8.45	3.43	8.95	3.21	8.83	3.27	64.611	0.000	0.057
	Clinical	12.10	3.51	11.03	2.80	11.27	2.99			
	Total	8.88	3.63	9.22	3.23	9.14	3.33	0.652	0.420	0.001
(SR)	Community	8.35	3.13	8.76	3.35	8.66	3.30	4.183	0.041	0.004
	Clinical	9.70	2.28	8.89	3.21	9.07	3.04			
	Total	8.51	3.07	8.78	3.33	8.72	3.27	0.302	0.583	0.000
(ED)	Community	7.63	2.91	8.75	3.07	8.48	3.07	72.025	0.000	0.063
	Clinical	11.10	3.07	10.91	2.76	10.96	2.82			
	Total	8.04	3.13	9.03	3.12	8.79	3.15	1.982	0.159	0.002

LA = logical analysis; PR = positive reappraisal; SG = seeking guidance and support; PS = problem solving; CA = cognitive avoidance; AR = acceptance or resignation; SR = seeking alternative rewards; ED = emotional discharge.

**Table 3 jcm-10-02499-t003:** Descriptive analysis for the four CRI-A dimensions and univariate analysis by group (community/clinical) and group-gender interaction.

CRI-A Dimensions	Sample	Male		Female		Total		Between-Subject Effects		
								Sample and Group-Gender Interaction		
		M	SD	M	SD	M	SD	F	*p*	*η*
Cognitive	Community	39.97	7.82	40.97	8.94	40.73	8.69	24.332	0.000	0.022
	Clinical	46.60	12.70	43.86	7.76	44.47	9.11			
	Total	40.75	8.77	41.34	8.85	41.20	8.83	2.773	0.082	0.004
Behavioural	Community	35.87	8.39	37.55	8.80	37.14	8.73	11.796	0.001	0.011
	Clinical	41.10	10.84	38.97	8.89	39.44	9.36			
	Total	36.48	8.86	37.73	8.82	37.43	8.84	3.858	0.050	0.004
Approach	Community	41.32	10.05	41.31	10.73	41.31	10.57	0.021	0.885	0.000
	Clinical	42.20	12.73	40.09	10.97	40.56	11.37			
	Total	41.42	10.38	41.15	10.76	41.22	10.67	0.795	0.373	0.001
Avoidance	Community	34.52	7.53	37.21	8.26	36.57	8.17	88.033	0.000	0.076
	Clinical	45.50	7.96	42.74	6.85	43.36	7.17			
	Total	35.81	8.35	37.92	8.30	37.42	8.35	9.599	0.002	0.009

**Table 4 jcm-10-02499-t004:** Repeated-measures analysis between the four CRI-A dimensions by group (community/clinical).

Sample	CRI-A Dimensions								Within-Subject Effects		
	Cognitive		Behavioural		Approach		Avoidance				
	*M*	*SD*	*M*	*SD*	*M*	*SD*	*M*	*SD*	*F*	*p*	*η*
Community	40.73	8.69	37.14	8.73	41.31	10.57	36.57	8.17	457.050	0.000	0.328
Clinical	44.47	9.11	39.44	9.36	40.56	11.37	43.36	7.17	5.140	0.025	0.037

**Table 5 jcm-10-02499-t005:** Sensitivity, specificity and Youden’s index for scores in the cognitive, behavioural and avoidance dimensions of the CRI-A to identify the presence of psychopathological disorders.

CRI-A	Cut-Off Point	Sensitivity	Specificity	Youden’s Index
Cognitive dimension	41.5 *	0.600	0.514	0.114
	42.0	0.589	0.548	0.137
	42.5 ***	0.578	0.581	0.159
	43.0 **	0.533	0.597	0.131
Behavioural dimension	37.5 *	0.556	0.524	0.081
	38.0	0.545	0.535	0.080
	38.5	0.533	0.546	0.079
	39.0	0.533	0.569	0.102
	39.5 ***	0.533	0.591	0.124
	40.0 **	0.511	0.610	0.121
Avoidance dimension	36.0 *	0.789	0.503	0.291
	37.5	0.733	0.524	0.257
	38.0	0.700	0.555	0.255
	38.5	0.667	0.585	0.252
	39.0	0.655	0.617	0.273
	39.5 ***	0.644	0.649	0.293
	40.0	0.622	0.662	0.284
	40.5	0.600	0.674	0.274
	41.0	0.589	0.687	0.276
	41.5	0.578	0.700	0.278
	42.0 **	0.533	0.726	0.259

* Score maximising sensitivity. ** Score maximising specificity. *** Score maximising sensitivity and specificity.

## Data Availability

Data is contained within the article.

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
