# Peer review of "Coping Strategies by University Students in Response to COVID-19: Differences between Community and Clinical Groups"

_jcm, 2021, doi:10.3390/jcm10112499_

Round 1
Reviewer 1 Report
In the paper "Coping Strategies by University Students in Response to COVID-19: Differences between Community and Clinical Groups", the authors present issues that are very important today, relating to ways of coping with the pandemic situation in the general population and persons in the clinical group. They show differences in coping styles and the dual struggle with the pandemic in the clinical group. The paper provides yet another contribution of researchers to the global discussion on the need for restrictions to protect human life and physical health while considering the need to protect mental health; isolation and various restrictions are a significant risk factor for that.
Strengths
The paper addresses important aspects of coping with the pandemic of individuals in the community and the clinical group.
The study involved a large sample of 1,074 students.
The results are clearly presented in tables and a figure.
The authors obtain some important results and formulate important conclusions; they underscore the need to monitor how people from different clinical groups cope with the pandemic situation, as well as the need to develop specific ways of helping.
The paper significantly contributes to international discussion and provides practical guidance for the area of health care.
Remarks
- The authors indicate that they used the reliability coefficient alpha. The text, however, lacks results showing the coefficient level in the method employed.
- The "Procedure" section should be extended. A detailed description of the clinical group is missing. How exactly were these subjects selected? What criterion was used and what are the characteristics of these persons? How can we be sure that they are a clinical group and what disorders do they suffer from? This description would benefit from expanding. It would be worth asking the question if the healthy and the clinical group are similar in terms of basic sociodemographic data and if the differences obtained follow merely from the fact of being qualified for the clinical group or from other variables, such as demographics (which we do not know).
- When exactly was this research conducted? Were the clinical and the non-clinical group examined simultaneously? Here, it might be useful to compare coping strategies also with the basic parameters regarding the pandemic (overall duration, duration of lockdown, number of COVID-19 cases and deaths during the pandemic – to be sure that specific strategies are activated as a result of pandemic restrictions). It seems that it would be useful to provide the exact timing of the study, possibly refine its scope, preferably in months/weeks rather than the years 2019/2020.

Author Response
Dear author:
We would like to thank you for considering our manuscript and for your carefully review. We are pleased to submit a revised version of our article, which has been substantially improved as a result of your comments and suggestions. We have tried to modify every aspect that has been pointed out (see changes in red). The manuscript also has been checked by a professional translator. We hope that you will consider this new version acceptable for its publication.
Authors appreciate all the recommendations made in order to improve the final manuscript.
We remain at your disposal for any questions you would have.
Sincerely,
The authors indicate that they used the reliability coefficient alpha. The text, however, lacks results showing the coefficient level in the method employed.
Answer: Cronbach’s alpha has been added to “Instruments”.
The "Procedure" section should be extended. A detailed description of the clinical group is missing. How exactly were these subjects selected? What criterion was used and what are the characteristics of these persons? How can we be sure that they are a clinical group and what disorders do they suffer from? This description would benefit from expanding. It would be worth asking the question if the healthy and the clinical group are similar in terms of basic sociodemographic data and if the differences obtained follow merely from the fact of being qualified for the clinical group or from other variables, such as demographics (which we do not know).
Answer: The clinical group’s composition has been detailed, as well as the origin and main characteristics of the individuals. Equivalence in clinical and community groups by gender, age and year has been included in “Participants”.
When exactly was this research conducted? Were the clinical and the non-clinical group examined simultaneously? Here, it might be useful to compare coping strategies also with the basic parameters regarding the pandemic (overall duration, duration of lockdown, number of COVID-19 cases and deaths during the pandemic – to be sure that specific strategies are activated as a result of pandemic restrictions). It seems that it would be useful to provide the exact timing of the study, possibly refine its scope, preferably in months/weeks rather than the years 2019/2020.
Answer: It has been specified that both groups answered the questionnaire at the same point and the precise moment of the study has been detailed. The situation in Spain at the time of the research has been described.
Finally, we would like to express our appreciation for your suggestion. Although it is not the main goal of the present study, we will take it into consideration for a second study in which we are evaluating the potential incidence of variables related to the specific situations in which participants experienced lockdown.
Reviewer 2 Report
General comments
The manuscript explores coping strategies in a large number of studies during the COVID-19 pandemic. This work is currently very important since school institutes around the world are searching for the best ways to guide their students, both high school and higher education, through the remainder of the lockdown and through re-entering once a lockdown is completed. This work is therefore important to identify groups needing more intensive guidance and to adjust current practices.
Specific comments
The manuscript as a whole is well written. I have three minor comments.
In the current form, the introduction is quite long and the structure is a bit unclear. For example lines 151-157 might be more suitable in section 2.2. And lines 157-174 might be combined with lines 90-101 instead of added at the end of the introduction.
Could you add the minimum and maximum scores per coping dimension and coping response to section 2.2. to make the results more easily interpretable for readers not familiar with the CRI-A?
Line 344-346, is that an assumption based on the diagnosis of the clinical group or did you measure this in your sample?
Author Response
Dear author:
We would like to thank you for considering our manuscript and for your carefully review. We are pleased to submit a revised version of our article, which has been substantially improved as a result of your comments and suggestions. We have tried to modify every aspect that has been pointed out (see changes in red). The manuscript also has been checked by a professional translator. We hope that you will consider this new version acceptable for its publication.
Authors appreciate all the recommendations made in order to improve the final manuscript.
We remain at your disposal for any questions you would have.
Sincerely,
In the current form, the introduction is quite long and the structure is a bit unclear. For example lines 151-157 might be more suitable in section 2.2. And lines 157-174 might be combined with lines 90-101 instead of added at the end of the introduction.
Answer: Review has been performed and the text has been restructured.
Could you add the minimum and maximum scores per coping dimension and coping response to section 2.2. to make the results more easily interpretable for readers not familiar with the CRI-A?
Answer: This has been added to “Instruments”.
Line 344-346, is that an assumption based on the diagnosis of the clinical group or did you measure this in your sample?
Answer: It is an assumption based on clinical population.
Reviewer 3 Report
Coping Strategies by University Students in Response to 
COVID-19: Differences between Community and Clinical Groups
The paper is well written and clear. However, there are some details that are not clear and are sometimes inconsistent that needs refining before it is suitable for publication..
Sample: It is stated that the sample size was calculated on the number of participants enrolled at the university, however, it is unclear on which outcome measure was it based and the expected effect size.
Also, it is unclear how were participants actually recruited and the completion rate.
Generalizability: how generalizable are the findings from your sample? please add in the limitation section.
Timing: when were participants recruited exactly?
How long have they been in lockdown when they answered?
Was the diagnosis of anxiety or depression done before the lockdown?
Statistical analyses: Did you apply any correction to p values for multiple comparisons?
I am not an expert in ROC curves, but isn’t a sensitivity and a specificity of 60% rather low, and not much more than chance?
Ethical approval: inconsistent information is provided: in the text it is not mentioned but in the end it is specified that it was approved by the Bioethics and Biosafety Committee of the University of Extremadura. 
Please add it into the text and add the protocol number.
Please clarify also how were participants contacted in the first place and how was their privacy respected while also respecting the described stratification? E.g., the link was published on university pages or participants were emailed directly.
Minor: please use the term individuals rather than subjects.
Minor: please add a table (maybe in the supplementary material) describing participants’ characteristics including age, sex, year of uni, course etc.. in the full sample and distinguishing the clinical and community sample.
Author Response
Dear author:
We would like to thank you for considering our manuscript and for your carefully review. We are pleased to submit a revised version of our article, which has been substantially improved as a result of your comments and suggestions. We have tried to modify every aspect that has been pointed out (see changes in red). The manuscript also has been checked by a professional translator. We hope that you will consider this new version acceptable for its publication.
Authors appreciate all the recommendations made in order to improve the final manuscript.
We remain at your disposal for any questions you would have.
Sincerely,
Sample: It is stated that the sample size was calculated on the number of participants enrolled at the university, however, it is unclear on which outcome measure was it based and the expected effect size.
We appreciate your suggestion and will take it into consideration for future samples. However, given the characteristics of our study (there are no PLACEBO groups, either control or treatment, intervention or technique) and the size of the sample, we did not think necessary to establish a minimum sample size.
Also, it is unclear how were participants actually recruited and the completion rate.
The students from the community group were not recruited, they were in lockdown at the time of doing the test. The individuals of the clinical group were reached following the register the Psychological Care Unit.
It has been added in the text.
Generalizability: how generalizable are the findings from your sample? please add in the limitation section.
It has been added in the text.
Timing: when were participants recruited exactly?
It is now detailed. From May 10 to June 10, 2020. It has been added in the text.
How long have they been in lockdown when they answered?
Participants had been in home lockdown for 64 days when they started the test (10 May 2020) and for 94 days when they finished it (10 June 2020).
Was the diagnosis of anxiety or depression done before the lockdown?
It is detailed. The individuals from the clinical group had been previously diagnosed with anxiety and/or depression disorders by both public and private professionals. It has been added to the text.
Statistical analyses: Did you apply any correction to p values for multiple comparisons?
Since the fixed factors are dichotomous variables, no corrections were required.
I am not an expert in ROC curves, but isn’t a sensitivity and a specificity of 60% rather low, and not much more than chance?
In clinical studies aiming to determine the efficiency of a drug or vaccine, a sensitivity and specificity of 60% would be low indeed. However, since our goal is to determine the discriminative accuracy of coping responses score, a sensitivity and specificity of 60% is significant.
Ethical approval: inconsistent information is provided: in the text it is not mentioned but in the end it is specified that it was approved by the Bioethics and Biosafety Committee of the University of Extremadura. 
Please add it into the text and add the protocol number.
It has been added in the text.
Please clarify also how were participants contacted in the first place and how was their privacy respected while also respecting the described stratification? E.g., the link was published on university pages or participants were emailed directly.
The evaluation test was a Google Forms document with no identification details whatsoever. It was distributed through several means amongst the students. As for the community group individuals, it was sent by e-mail from different university centres; it was also uploaded to the Virtual Campus thanks to the collaboration of a great deal of lecturers. As for the clinical group individuals, they received a personalised e-mail containing the link to Google Forms, since they were registered in our Psychological Care Unit.
Minor: please use the term individuals rather than subjects.
It has been modified.
Minor: please add a table (maybe in the supplementary material) describing participants’ characteristics including age, sex, year of uni, course, etc. in the full sample and distinguishing the clinical and community sample.
We do not think necessary to duplicate this information because there are characteristics of the participants that are not relevant for the study. Other data are clearly identified in “Participants”.